# Factorized Diffusion Architectures for Unsupervised Image Generation and Segmentation

**Xin Yuan**[†‡*]
[†]Google
yuanxzzz@google.com

**Michael Maire**[‡]
[‡]University of Chicago
mmaire@uchicago.edu

## Abstract

We develop a neural network architecture which, trained in an unsupervised manner as a denoising diffusion model, simultaneously learns to both generate and segment images. Learning is driven entirely by the denoising diffusion objective, without any annotation or prior knowledge about regions during training. A computational bottleneck, built into the neural architecture, encourages the denoising network to partition an input into regions, denoise them in parallel, and combine the results. Our trained model generates both synthetic images and, by simple examination of its internal predicted partitions, semantic segmentations of those images. Without fine-tuning, we directly apply our unsupervised model to the downstream task of segmenting real images via noising and subsequently denoising them. Experiments demonstrate that our model achieves accurate unsupervised image segmentation and high-quality synthetic image generation across multiple datasets.

## 1 Introduction

Supervised deep learning yields powerful discriminative representations, and has fundamentally advanced many computer vision tasks, including image classification [13, 58, 21, 28], object detection [18, 49, 41], and semantic and instance segmentation [42, 22, 33]. Yet, annotation efforts [13], especially those involving fine-grained labeling for tasks such as segmentation [39], can become prohibitively expensive to scale with increasing dataset size. This motivates an ongoing revolution in self-supervised methods for visual representation learning, which do not require any annotated data during a large-scale pre-training phase [7, 15, 67, 35, 23, 10, 12]. However, many of these approaches, including those in the particularly successful contrastive learning paradigm [23, 10, 12], still require supervised fine-tuning (*e.g.,* linear probing) on labeled data to adapt networks to downstream tasks such as classification [23, 10] or segmentation [8, 69].

In parallel with the development of self-supervised deep learning, rapid progress on a variety of frameworks for deep generative models [32, 19, 65, 66, 60, 37, 27, 59, 50] has lead to new systems for high-quality image synthesis. This progress inspires efforts to explore representation learning within generative models, with recent results suggesting that image generation can serve as a good proxy task for capturing high-level semantic information, while also enabling realistic image synthesis.

Building upon generative adversarial networks (GANs) [19] or variational autoencoders (VAEs) [32], InfoGAN [11] and Deep InfoMax [26] demonstrate that generative models can perform image classification without any supervision. PerturbGAN [4] focuses on a more complex task, unsupervised image segmentation, by forcing an encoder to map an image to the input of a pre-trained generator so that it synthesizes a composite image that matches the original input image. However, here training is conducted in two stages and mask generation relies on knowledge of predefined object classes.

---

[*]This work was completed while Xin Yuan was a PhD student at the University of Chicago.

38th Conference on Neural Information Processing Systems (NeurIPS 2024).

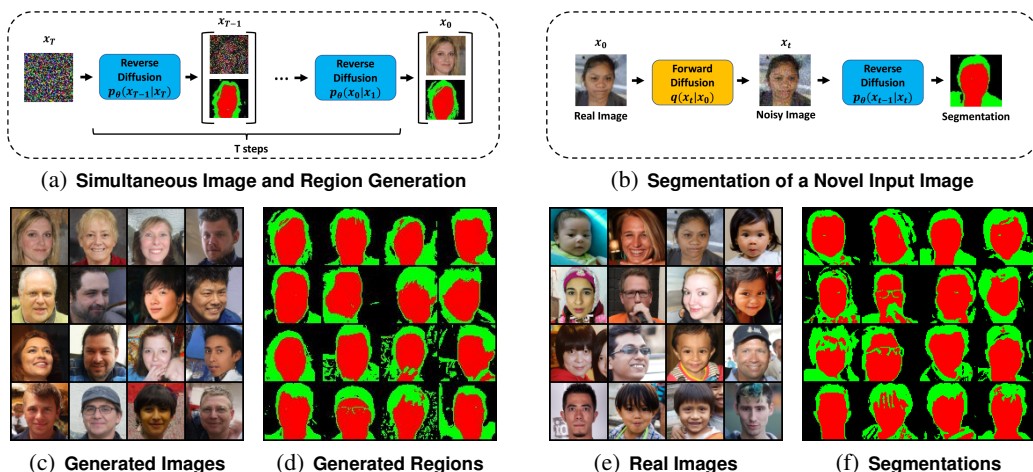

Figure 1: **Unifying image generation and segmentation.** (a) We design a denoising diffusion model with a specific architecture that couples region prediction with spatially-masked diffusion over predicted regions, thereby generating both simultaneously. (b) An additional byproduct of running our trained denoising model on an arbitrary input image is a segmentation of that image. Using a model trained on FFHQ [31], we achieve both high quality synthesis of images and corresponding semantic segmentations (c-d), as well as the ability to accurately segment images of real faces (e-f). Segmenting a real image is fast, requiring only one forward pass (one denoising step).

Denoising diffusion probabilistic models (DDPMs) [27] also achieve impressive performance in generating realistic images. DatasetDDPM [2] investigates the intermediate activations from the pre-trained U-Net [51] network that approximates the Markov step of the reverse diffusion process in DDPM, and proposes a simple semantic segmentation pipeline fine-tuned on a few labeled images. In spite of this usage of labels, DatasetDDPM demonstrates that high-level semantic information, which is valuable for downstream vision tasks, can be extracted from pre-trained DDPM U-Net. Diff-AE [48] and PADE [70] are recently proposed methods for representation learning by reconstructing images in the DDPM framework. However, their learned representations are in the form of a latent vector containing information applicable for image classification.

In contrast to all of these methods, we demonstrate a fundamentally new paradigm for unsupervised visual representation learning with generative models: constrain the architecture of the model with a structured bottleneck that provides an interpretable view of the generation process, and from which one can simply read off desired latent information. This structured bottleneck does not exist in isolation, but rather is co-designed alongside the network architecture preceding and following it. The computational layout of these pieces must work together in a manner that forces the network, when trained from scratch for generation alone, to populate the bottleneck data structure with an interpretable visual representation.

We demonstrate this concept in the scenario of a DDPM for image generation and semantic segmentation as the interpretable representation to be read from the bottleneck. Thus, we frame unsupervised image segmentation and generation in a unified system. Moreover, experiments demonstrate that domain-specific bottleneck design not only allows us to accomplish an end task (segmentation) for free, but also boosts the quality of generated samples. This challenges the assumption that generic architectures (*e.g.,* Transformers [61]) alone suffice; we find synergy by organizing such generic building blocks into a factorized architecture which generates different image regions in parallel.

Figure 1 provides an overview of our setting alongside example results, while Figure 2 illustrates the details of our DDPM architecture which are fully presented in Section 3. This architecture constrains the computational resources available for denoising in a manner that encourages learning of a factorized model of the data. Specifically, each step of the DDPM has the ability to utilize additional inference passes through multiple copies of a subnetwork if it is willing to decompose the denoising task into parallel subproblems. The specific decomposition strategy itself must be learned, but, by design, is structured in a manner that reveals the solution to our target task of image segmentation. We summarize our contributions as three-fold:

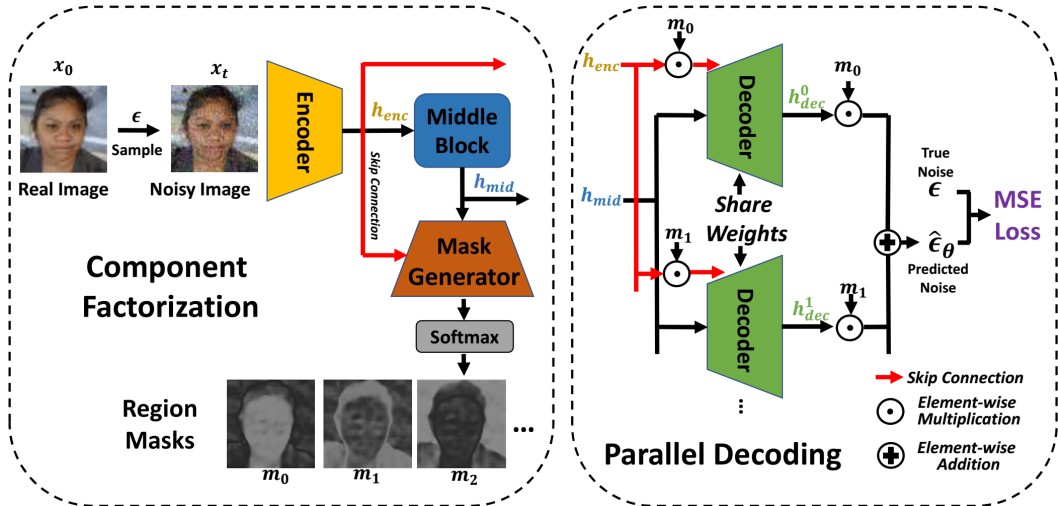

Figure 2: **Factorized diffusion architecture.** Our framework restructures the architecture of the neural network within a DDPM [27] so as to decompose the image denoising task into parallel subtasks. All modules are end-to-end trainable and optimized according to the same denoising objective as DDPM. *Left: Component factorization.* An *Encoder*, equivalent to the first half of a standard DDPM U-Net architecture, extracts features $h_{enc}$. A common *Middle Block* processes *Encoder* output into shared latent features $h_{mid}$. Note that *Middle Block* and $h_{mid}$ exist in the standard denoising DDPM U-Net by default. We draw it as a standalone module for a better illustration of the detailed architectural design. A *Mask Generator*, structured as the second half of a standard U-Net receives $h_{mid}$ as input, alongside all encoder features $h_{enc}$ injected via skip connections to layers of corresponding resolution. This later network produces a soft classification of every pixel into one of $K$ region masks, $m_0, m_1, ..., m_K$. *Right: Parallel decoding.* A *Decoder*, also structured as the second half of a standard U-Net, runs separately for each region. Each instance of the *Decoder* receives shared features $h_{mid}$ and a masked view of encoder features $h_{enc} \odot m_i$ injected via skip connections to corresponding layers. Decoder outputs are masked prior to combination. Though not pictured, we inject timestep embedding $t$ into the *Encoder*, *Mask Generator*, and *Decoder*.

- **Unified learning of generation and segmentation.** We train our new DDPM architecture once, obtaining a model directly applicable to two different tasks with zero modification or fine-tuning: image generation and image segmentation. Segmenting a novel input image is fast, comparable in speed to any system using a single forward pass of a U-Net [51] like architecture.
- **Unsupervised segmentation for free.** Our method automatically learns meaningful regions (*e.g.,* foreground and background), guided only by the DDPM denoising objective; no extra regularization terms, no use of labels.
- **Higher quality image synthesis.** Our model generates higher-quality images than the baseline DDPM, as well as their corresponding segmentations simultaneously. We achieve excellent quantitative and qualitative results under common evaluation protocols (Section 4).

Beyond improvements to image generation and segmentation, our work is a case study of a new paradigm for using generation as a learning objective, in combination with model architecture as a constraint. Rather than viewing a pre-trained generative model as a source from which to extract and repurpose features for downstream tasks, design the model architecture in the first place so that, as a byproduct of training from scratch to generate, it also learns to perform the desired task.

## 2 Related Work

**Image Segmentation.** Generic segmentation, which seeks to partition an image into meaningful regions without prior knowledge about object categories present in the scene, is a longstanding challenge for computer vision. Early methods rely on combinations of hand-crafted features based on intensity, color, and texture cues [6, 44], clustering algorithms [57], and a duality between closed

contours and the regions they bound [1]. Deep learning modernized the feature representations used in these pipelines, yielding systems which, trained with supervision from annotated regions [43], reach near human-level accuracy on predicting and localizing region boundaries [3, 56, 64, 34].

Semantic segmentation, which assigns a category label to each pixel location in image, has been similarly revolutionized by deep learning. Here, the development of specific architectures [42, 51, 20] enabled porting of approaches for image classification to the task of semantic segmentation.

Recent research has refocused on the challenge of learning to segment without reliance on detailed annotation for training. Hwang et al. [29] combine two sequential clustering modules for both pixel-level and segment-level to perform this task. Ji et al. [30] and Ouali et al. [46] follow the concept of mutual information maximization to partition pixels into two segments. Savarese et al. [54] further propose a learning-free adversarial method from the information theoretic perspective, with the goal of minimizing predictability among different pixel subsets. Note that even completely unsupervised foreground/background segmentation is a non-trivial task. Liu et al. [40], a recent advance in this regime, produces similar region mask output, yet depends entirely upon motion cues from video for training. We achieve such unsupervised learning from static images alone.

**Learning Segmentation in Generative Models.** Previous generative model-based approaches learn semantic segmentation by perturbing [4] or redrawing [9] generated foreground and background masks. Despite good performance, these methods apply only to two-class partitions and require extra loss terms based upon object priors in training datasets.

Denoising diffusion probabilistic models (DDPMs) [27] achieve state-of-the-art performance in generating realistic images. Their noise schedule in training may offer advantages for scaling up models in a stable manner. Recent works [2, 48, 70] explore representation learning capability in DDPMs. DatasetDDPM [2] examines few-shot segmentation with pre-trained diffusion models, but requires human labels to train a linear classifier. With the default U-Net architecture [51], it loses the efficiency and flexibility of generating image and masks in a single-stage manner. Diff-AE [48] and PADE [70] perform representation learning driven by a reconstruction objective in the DDPM framework. Unfortunately, their learned latent vectors are not applicable to more challenging segmentation tasks and they require a pre-trained interpreter to perform downstream image classification.

DiffuMask [63] takes a pre-trained Stable Diffusion model [50], which is built using large-scale text-to-image datasets (and thus solves a far less challenging problem), and conducts a post-hoc investigation on how to extract segmentation from its attention maps. Neither our system, nor the baseline DDPM to which we compare, makes use of such additional information. Furthermore, DiffuMask does not directly output segmentation; it is basically a dataset generator, which produces generated images and pseudo labels, which are subsequently used to train a separate segmentation model. Our method, in contrast, is both completely unsupervised and provides an end-to-end solution by specifying an architectural design in which training to generate reveals segmentations as a bonus.

MAGE [38] shares with us a similar motivation of framing generation and representation learning in a unified framework. However, our approach is distinct in terms of both (1) task: we tackle a more complex unsupervised segmentation task (without fine-tuning) instead of image classification (with downstream fine-tuning), and (2) design: 'masks' play a fundamentally different role in our system. MAGE adopts an MAE [24]-like masking scheme on input data, in order to provide a proxy reconstruction objective for self-supervised representation learning. Our use of region masks serves a different purpose, as they are integral components of the model being learned and facilitate factorization of the image generation process into parallel synthesis of different segments.

BlobGAN [16] is a generative model for creating images with fine-grained control over the spatial arrangement of content. It leverages blob-like components instead of accurate region masks as basic building blocks for the synthesis process, allowing for intuitive content manipulation. In the generative modeling space, BlobGAN serves a different purpose than our method: BlobGAN excels in scenarios requiring explicit spatial control and interactive editing, while our factorized diffusion approach provides a framework for learning high-quality image generation and segmentation.

## 3 Factorized Diffusion Models

Figure 2 illustrates the overall architecture of our system, which partitions the denoising network within a diffusion model into an unsupervised region mask generator and parallel per-region decoders.

## 3.1 Unsupervised Region Factorization

To simultaneously learn representations for both image generation and unsupervised segmentation, we first design the region mask generator based on the first half (encoder) of a standard DDPM U-Net. We obtain input $\boldsymbol{x}_t$, a noised version of $\boldsymbol{x}_0$, via forward diffusion:

$$q(\boldsymbol{x}_t|\boldsymbol{x}_0) := \mathcal{N}(\boldsymbol{x}_t; \sqrt{\bar{\alpha}_t}\boldsymbol{x}_0, (1-\bar{\alpha}_t)I),$$
$$\boldsymbol{x}_t = \sqrt{\bar{\alpha}_t}\boldsymbol{x}_0 + \sqrt{1-\bar{\alpha}_t}\boldsymbol{\epsilon}, \boldsymbol{\epsilon} \sim \mathcal{N}(0,1), \tag{1}$$

where $\alpha_t = 1 - \beta_t, \bar{\alpha}_t = \prod_{s=1}^{t} \alpha_t$.

In addition to the encoder half of the U-Net, we instantiate a middle block consisting of layers operating on lower spatial resolution features. Parameterizing these subnetworks as $\theta_{enc}$ and $\theta_{mid}$, we extract latent representations:

$$\boldsymbol{h}_{enc} = \theta_{enc}(\boldsymbol{x}_t, t), \tag{2}$$
$$\boldsymbol{h}_{mid} = \theta_{mid}(\boldsymbol{h}_{enc}, t) \tag{3}$$

where $\boldsymbol{h}_{enc}$ encapsulates features at all internal layers of $\theta_{enc}$, for subsequent use as inputs, via skip connections, to corresponding layers of decoder-style networks (second half of a standard U-Net).

We instantiate a mask generator, $\theta_{mask}$, as one such decoder-style subnetwork. A softmax layer produces an output tensor with $K$ channels, representing $K$ different regions in image $\boldsymbol{x}_0$:

$$\boldsymbol{m}_k = \theta_{mask}(\boldsymbol{h}_{mid}, \boldsymbol{h}_{enc}, t) \tag{4}$$

Following a U-Net architecture, $\boldsymbol{h}_{enc}$ feeds into $\theta_{mask}$ through skip-connections.

## 3.2 Parallel Decoding Through Weight Sharing

We aim to extend a standard DDPM U-Net decoder $\theta_{dec}$ to consider region structure during generation. One simple design is to condition on $\boldsymbol{m} = \{\boldsymbol{m_0}, \boldsymbol{m_1}, ...\}$ by concatenating it with input $\boldsymbol{h}_{mid}$ and $\boldsymbol{h}_{enc}$ along the channel dimension:

$$\hat{\boldsymbol{\epsilon}} = \theta_{dec}(\text{concat}[\boldsymbol{h}_{mid}, \boldsymbol{m}], \text{concat}[\boldsymbol{h}_{enc}, \boldsymbol{m}], t), \tag{5}$$

where $\boldsymbol{h}_{mid}$ and $\boldsymbol{h}_{enc}$ are generated from Eq. 2 and Eq. 3. We downsample $\boldsymbol{m}$ accordingly to the same resolution as $\boldsymbol{h}_{mid}$ and $\boldsymbol{h}_{enc}$ at different stages. However, such a design significantly modifies (*e.g.,* channel sizes) the original U-Net decoder architecture. Moreover, conditioning with the whole mask representation may also result in a trivial solution that simply ignores region masks.

To address these issues, we separate the decoding scheme into multiple parallel branches of weight-shared U-Net decoders, each masked by a single segment. Noise prediction for $k$-th branch is:

$$\hat{\boldsymbol{\epsilon}}_k = \theta_{dec}(\boldsymbol{h}_{mid}, \boldsymbol{h}_{enc} \odot \boldsymbol{m}_k, t) \tag{6}$$

and the output is a sum of region-masked predictions:

$$\hat{\boldsymbol{\epsilon}} = \sum_{k=0}^{K-1} \hat{\boldsymbol{\epsilon}}_k \odot \boldsymbol{m}_k \tag{7}$$

## 3.3 Optimization with Denoising Objective

We train our model in an end-to-end manner, driven by the simple DDPM denoising objective. Model weights $\theta = \{\theta_{enc}, \theta_{mid}, \theta_{dec}, \theta_{mask}\}$ are optimized by minimizing the noise prediction loss:

$$L = \mathbb{E}||\boldsymbol{\epsilon} - \hat{\boldsymbol{\epsilon}}||_2^2 \tag{8}$$

Unlike previous work, our method does not require a mask regularization loss term [54, 4, 9], which predefines mask priors (*e.g.,* object size). Algorithm 1 summarizes training.

### 3.4 Segmentation via Reverse Diffusion

Once trained, our model can both segment novel input images and synthesize images from noise.

**Real Image Segmentation.** Given clean input image $x_0$, we first sample a noisy version $x_t$ through forward diffusion in Eq. 1. We then perform one-step denoising by passing $x_t$ to the model. We collect the predicted region masks as the segmentation for $x_0$ using Eq. 4.

**Image and Mask Generation.** Using reverse diffusion, our model can generate realistic images and their corresponding segmentation masks, starting from a pure noise input $x_T \sim \mathcal{N}(0, 1)$. Reverse diffusion predicts $x_{t-1}$ from $x_t$:

$$x_{t-1} = 1/\sqrt{\alpha_t}(x_t - \frac{1 - \alpha_t}{\sqrt{1 - \bar{\alpha}_t}}\theta(x_t, t)) + \sigma_t z, \tag{9}$$

$$z \sim \mathcal{N}(0, 1) \quad \text{if} \quad t > 1 \quad \text{else} \quad z = 0. \tag{10}$$

where $\sigma_t$ is empirically set according to the DDPM noise scheduler. We perform $T$ steps of reverse diffusion to generate an image. We also collect its corresponding masks using Eq. 4 when $t = 1$. Algorithm 2 summarizes this process.

| **Algorithm 1** | **Algorithm 2** |
|---|---|
| Training Masked Diffusion | Image and Mask Generation |
| **Input:** Data $x_0$ | **Input:** Noise $x_T$, trained model $\theta$ |
| **Output:** Trained model $\theta$ | **Output:** |
| **Initialize:** |     Image $\hat{x}_0$ and segmentation $\hat{m}_0$ |
|     Model weights $\theta$, Timesteps T | **Initialize:** $x_T \sim \mathcal{N}(0, 1)$ |
| **for** iter $= 1$ **to** Iter$_{total}$ **do** | **for** t $= T$ **to** 1 **do** |
|     Sample $t \in [1, T]$ |     Sample $z$ using Eq. 10 |
|     Sample $x_t$ using Eq. 1 |     Perform reverse diffusion using Eq. 9 |
|     Calculate $\hat{\epsilon}$ using Eq. 7 |     **if** $t = 1$ **then** |
|     Backprop with Eq. 8 |         Collect $\hat{m}_0$ using Eq. 4 |
|     Update $\theta$ |         Return $\hat{x}_0$ and $\hat{m}_0$ |
| **end for** |     **end if** |
| return $\theta$ | **end for** |

## 4 Experiments

We evaluate on: (1) real image segmentation, (2) image and region mask generation, using Flower [45], CUB [62], FFHQ [31], CelebAMask-HQ [36], and ImageNet [53]. In addition to the design of flat set of $K$ regions, we also conduct a preliminary investigation into reorganizing our architectural design to support hierarchical segmentations; see Section A.1.

**Evaluation Metrics.** For unsupervised segmentation on Flower and CUB, we follow the data splitting in IEM [54] and evaluate predicted mask quality under three commonly used metrics, denoted as Acc., IOU and DICE score [54, 9]. Acc. is the (per-pixel) mean accuracy of the foreground prediction. IOU is the predicted foreground region's intersection over union with the ground-truth foreground region. DICE score is defined as $2\frac{\hat{F} \cap F}{|\hat{F}|}$ [14]. On ImageNet, we evaluate our method on Pixel-ImageNet [68], which provides human-labeled segmentation masks for 0.485M images covering 946 object classes. We report Acc., IOU and DICE score on a randomly sampled subset, each class containing at most 20 images. For face datasets, we train our model on FFHQ and only report per-pixel accuracy on the CelebAMask test set, using provided ground-truth.

For image and mask generation, we use Fréchet Inception Distance (FID) [25] for generation quality assessment. Since we can not obtain the ground-truth for generated masks, we apply a supervised U-Net segmentation model, pre-trained on respective datasets, to the generated images and measure the consistency between masks in terms of per-pixel accuracy. In addition to quantitative comparisons, we show extensive qualitative results.

**Implementation Details.** We train Flower, CUB and Face models at both $64 \times 64$ and $128 \times 128$ resolution. We also train class-conditioned ImageNet models with $64 \times 64$ resolution. For all

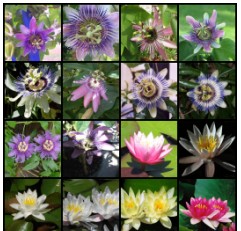 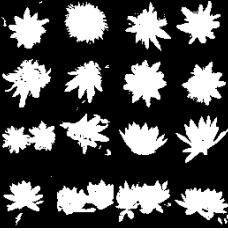

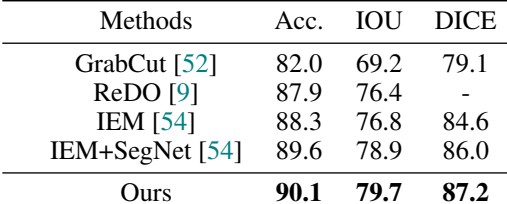

| Methods | Acc. | IOU | DICE |
|---|---|---|---|
| GrabCut [52] | 82.0 | 69.2 | 79.1 |
| ReDO [9] | 87.9 | 76.4 | - |
| IEM [54] | 88.3 | 76.8 | 84.6 |
| IEM+SegNet [54] | 89.6 | 78.9 | 86.0 |
| Ours | **90.1** | **79.7** | **87.2** |

(a) **Real Images**    (b) **Segmentation**

Figure 3: Segmentation on Flower.

Table 1: Comparisons on Flower.

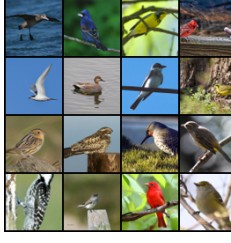 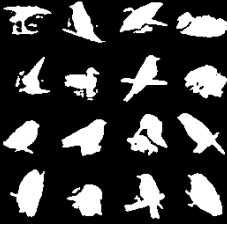

| Methods | Acc. | IOU | DICE |
|---|---|---|---|
| GrabCut [52] | 72.3 | 36.0 | 48.7 |
| PerturbGAN [4] | - | 38.0 | - |
| ReDO [9] | 84.5 | 42.6 | - |
| IEM [54] | 88.6 | 52.2 | 66.0 |
| IEM+SegNet [54] | 89.3 | 55.1 | 68.7 |
| Ours | **89.6** | **56.1** | **69.4** |

(a) **Real Images**    (b) **Segmentation**

Figure 4: Segmentation on CUB.

Table 2: Comparisons on CUB.

experiments, we use the U-Net [51] encoder-middle-decoder architecture similar to [27]. We use the decoder architecture as our mask generator and set the number of factorized masks $K$ as 3. We note that $K$ is the maximum number of regions the model may use. It could learn fewer components during training. For binary segmentation, we found setting $K = 3$ rather than $K = 2$ to assist training, with learned regions emerging as foreground, background, and a contour or transition between the two. For segmentation evaluation, we simply select the mask channel that emerges as foreground and apply standard benchmarks. For $64 \times 64$ the architecture is as follows: The downsampling stack performs four steps of downsampling, each with 3 residual blocks. The upsampling stack is setup as a mirror image of the downsampling stack. From highest to lowest resolution, U-Net stages use $[C, 2C, 3C, 4C]$ channels, respectively. For $128 \times 128$ architecture, the down/up sampling block is 5-step with $[C, C, 2C, 3C, 4C]$ channels, each with two residual blocks, respectively. We set $C = 128$ for all models.

We use Adam to train all the models with a learning rate of $10^{-4}$ and an exponential moving average (EMA) over model parameters with rate 0.9999. For all datasets except ImageNet, we train $64 \times 64$ and $128 \times 128$ models on 8 and 32 Nvidia V100 32GB GPUs, respectively. For Flower, CUB and FFHQ, we train the models for 50K, 50K, 500K iterations with batch size of 128, respectively. For ImageNet, we train 500K iterations on 32 Nvidia V100 GPUs with batch size 512. We adopt the linear noise scheduler as in Ho *et al.* [27] with $T = 1000$ timesteps.

## 4.1 Image Segmentation

To evaluate our method on real image segmentation, we set $t$ as 30 for the forward diffusion process. We also investigate the segmentation results with different noise levels in Figure 18. For Flower and CUB, Figures 3 and 4 show test images and predicted segmentations. Tables 1 and 2 provide quantitative comparison with representative unsupervised image segmentation methods: GrabCut [52], ReDO [9] and IEM [54]. As shown in Table 1 and Table 2, our method outperforms all competitors.

We also visualize the predicted face parsing results on FFHQ and CelebAMask datasets in Figure 1(c)(d) and Figure 5. Our model learns to accurately predict three segments corresponding to semantic components: skin, hair, and background. This particular semantic partitioning emerges from our unsupervised learning objective, without any additional prior. With ground-truth provided on CelebAMask-HQ, we also compare the pixel accuracy and mean IOU with a supervised U-Net and DatasetDDPM [2]. For the former, we train a supervised segmentation model with 3-class cross-entropy loss. For the unsupervised setting, we perform K-means (K=3) on the pre-trained DDPM, denoted as DatasetDDPM-unsup. Table 3 shows that we outperform DatasetDDPM by a large margin and achieve a relatively small performance gap with a supervised U-Net.

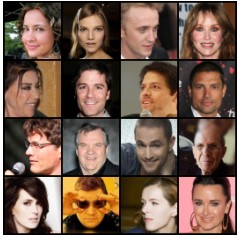 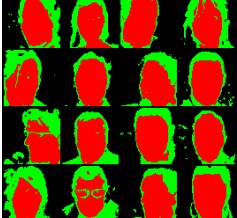

| Methods | Acc. | mIOU |
|---|---|---|
| Supervised UNet | 95.7 | 90.2 |
| DatasetDDPM-unsup. [2] | 78.5 | 69.3 |
| Ours | 87.9 | 80.3 |

Table 3: Seg. comparisons on CelebA.

(a) **Real Images**      (b) **Segmentation**

Figure 5: Segmentation on CelebA.

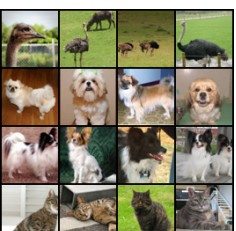 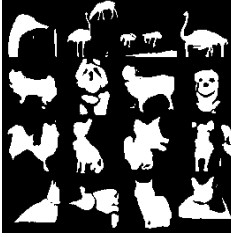

| Methods | Acc. | mIOU |
|---|---|---|
| Supervised UNet | 85.7 | 74.1 |
| DatasetDDPM-unsup. [2] | 74.1 | 60.4 |
| Ours | 80.7 | 67.7 |

Table 4: Seg. comparisons on ImageNet.

(a) **Real Images**      (b) **Segmentation**

Figure 6: Segmentation on ImageNet.

Figure 6 shows the accurate segmentation results for ImageNet classes: ostrich, pekinese, papillon, and tabby. We compare with supervised U-Net and DatasetDDPM-unsup in Table 4. We show more visualizations in the Appendix.

## 4.2 Image and Mask Generation

We evaluate our method on image and mask generation. As shown in Figure 7, 8, 1(c)(d) and 9, our method is able to generate realistic images. In the upper row of Table 5, we see a consistent quality improvement over the original DDPM. This suggests that our method is a better generation architecture than the standard U-Net; separate computational paths for denoising individual image regions is a beneficial prior to impose when learning to model the image distribution. Additionally, our method produces accurate corresponding masks, closely aligned with the semantic partitions in the generated image.

We also evaluate the quality of these segmentations. Since there is no ground-truth mask provided for generated images, we apply the U-Net segmentation models (pre-trained on respective labeled training sets) to the generated images to produce reference masks. We measure the consistency between the reference and the predicted parsing results in terms of pixel-wise accuracy. We compare our method with a pre-trained DDPM baseline, in which we first perform image generation, then pass them to DatasetDDPM-unsup to get masks. As shown in Table 5 (bottom), our method consistently achieves better segmentation on generated images than the DDPM baseline. Note that, different from the two-stage baseline, our method performs the computation in a single stage, generating image and mask simultaneously. The Appendix shows more visualizations.

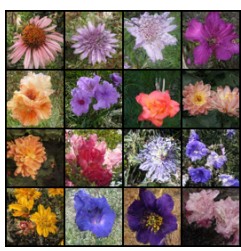 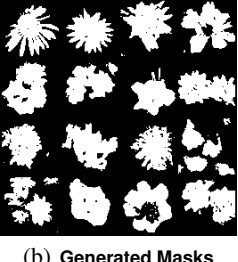      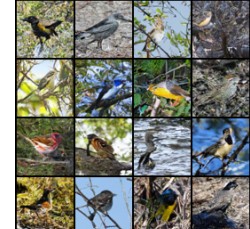 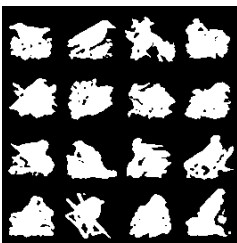

(a) **Generated Images**    (b) **Generated Masks**       (a) **Generated Images**    (b) **Generated Masks**

Figure 7: Generation on Flower.          Figure 8: Generation on CUB.

Table 5: Image and mask generation comparison on all datasets (top: FID(↓) bottom: Acc. (↑)).

| Models | Flower-64 | Flower-128 | CUB-64 | CUB-128 | FFHQ-64 | FFHQ-128 | ImageNet-64 |
|--------|-----------|------------|--------|---------|---------|----------|-------------|
| DDPM | 15.81 | 14.62 | 14.45 | 14.01 | 13.72 | 13.35 | 7.02 |
| Ours | **13.33** | **11.50** | **10.91** | **10.28** | **12.02** | **10.79** | **6.54** |
| DDPM | 80.5 | 82.9 | 84.2 | 83.7 | 84.2 | 84.2 | 71.2 |
| Ours | **92.3** | **92.7** | **91.4** | **91.2** | **90.3** | **90.7** | **84.1** |

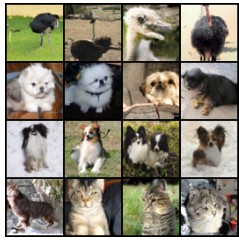 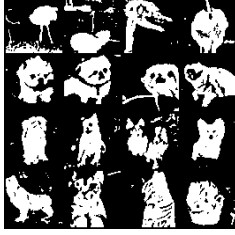

(a) **Generated Images**   (b) **Generated Masks**

Figure 9: Conditional generation on ImageNet.

| Methods | IOU.(↑) | FID (↓) |
|---------|---------|---------|
| Concat | 20.7 | 14.21 |
| Masking $h_{mid}$ | 20.2 | 14.33 |
| w/o weight sharing | 50.5 | 17.21 |
| Ours | **56.1** | **10.28** |

Table 6: Ablations of decoding scheme on CUB.

## 4.3 Ablation Study and Analysis

**Multi-branch Decoders with Weight Sharing.** Separating computation in multi-branch decoders with weight sharing is an essential design in our method. We show the effectiveness of this design by varying how to apply factorized masks in our decoding scheme: (1) concat: we use single branch to take concatenation of $h$ and $m$. (2) masking $h_{mid}$: we use $m$ to mask $h_{mid}$ instead of $h_{enc}$. (3) w/o weight sharing: we train decoders separately in our design. Table 6 shows separate design consistently yields better visual features than other designs for CUB. This suggests that our design benefits from fully utilizing mask information in the end-to-end denoising task and avoids a trivial solution where masks are simply ignored.

**Investigation on Mask Factorization.** Our architecture is able to generate factorized representations, each representing a particular segment of the input image. We show this by visualizing the individual channels from softmax layer output in our mask generator. As shown in Figure 10, skin, hair, and background are separated in different channels.

**Mask Refinement along Diffusion Process.** In the DDPM Markov process, the model implicitly formulates a mapping between noise and data distributions. We validate that this occurs for both images and latent region masks by visualizing image and mask generation along the sequential reverse diffusion process in Figure 11. We observe gradual refinement as denoising steps approach $t = 0$.

**Zero-shot Object Segmentation.** We evaluate zero-shot object segmentation on both PASCAL VOC 2012 [17] and DAVIS-2017 videos [47]. Baseline DDPM generation is not solved for these datasets when training from scratch without external large-scale datasets (*e.g.,* LAION [55] used in Stable Diffusion [50]). We directly adopt zero-shot transfer from our pre-trained ImageNet model by applying the conditional label mapping to VOC. Table 7 details the mapping rule. Figure 13 shows the accurate segmentation results for images of classes: aeroplane, monitor, person, and sofa from VOC. Since our method does not require any pixel labels, we evaluate the performance of

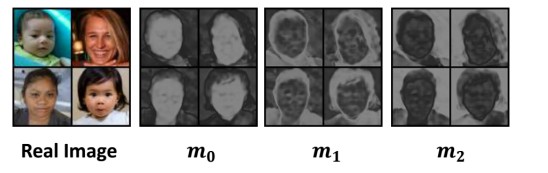

Real Image   $m_0$   $m_1$   $m_2$

Figure 10: Mask factorization (3 parts) on FFHQ.

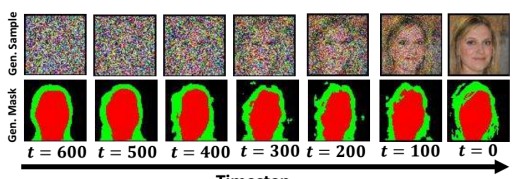

$t = 600$  $t = 500$  $t = 400$  $t = 300$  $t = 200$  $t = 100$  $t = 0$

Timestep

Figure 11: Gen. refinement along diffusion.

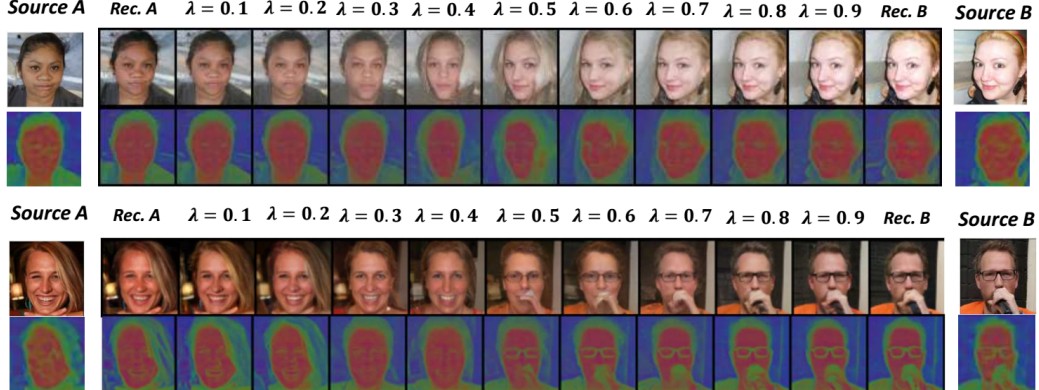

Figure 12: Interpolations on FFHQ with 250 timesteps of diffusion.

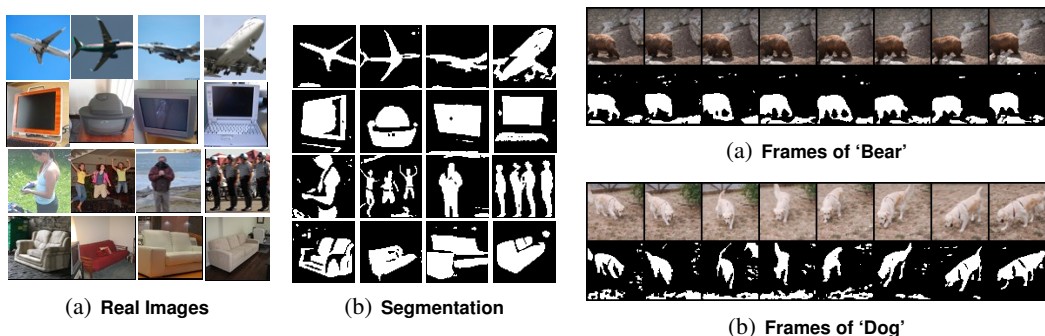

(a) **Real Images**    (b) **Segmentation**

Figure 13: Segmentation on VOC-2012.

(a) **Frames of 'Bear'**

(b) **Frames of 'Dog'**

Figure 14: Segmentation on DAVIS-17.

each object class individually. We report pixel accuracy and mIOU of each class in VOC in Table 7, which demonstrates that our method can achieve reasonably high performance. Our method achieves an accuracy of **0.78** and mIOU of **0.54** when averaging over all 20 classes. We also show video segmentation on DAVIS-2017 in Figure 14 and the Appendix, without any labeled video pre-training.

**Face Interpolation.** We also investigate face interpolation on FFHQ. Similar to standard DDPM [27], we perform the interpolation in the denoising latent space with 250 timesteps of diffusion. Figure 12 shows good reconstruction in both pixels and region masks, yielding smoothly varying interpolations across face attributes such as pose, skin, hair, expression, and background.

## 5   Conclusion

We propose a factorized architecture for diffusion models that is able to perform unsupervised image segmentation and generation simultaneously, while being trained once, from scratch, for image generation via denoising alone. Using model architecture as a constraint, via carefully designed component factorization and parallel decoding schemes, our method effectively and efficiently bridges these two challenging tasks in a unified framework, without the need for fine-tuning or altering the original DDPM training objective. Our work is the first example of engineering an architectural bottleneck so that learning a desired end task becomes a necessary byproduct of training to generate.

Our work is at the stage of a new architectural design for diffusion-based segmentation and generation, with 2- or 3-class segmentation results demonstrating improvements across multiple datasets, scaling up to ImageNet. Our initial investigation into hierarchical extensions suggests a promising future path towards handling complex scenes.

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

# A   Appendix

## A.1   Hierarchical Factorized Diffusion

We conduct a further investigation to reorganize our architectural design to support hierarchical mask factorization in place of a flat set of $K$ regions. We formulate a hierarchical factorized diffusion architecture to progressively refine segmentation results from a coarse initial prediction to a fine, detailed final segmentation. This approach helps in capturing both global context and fine details in the segmentation task. As shown in Figure 15, the first level replicates the factorized diffusion architecture depicted in Figure 2 to generate initial region masks $m_0^0, m_1^0, ...$, each applied on the noisy input for the next level factorized diffusion process. Each branch of the second level architecture generates finer representations of region masks $m_0^1, m_1^1, ...$, constructing the final denoising output as $\sum_i m_i^0 \frac{(h_i^0 + \sum_j h_j^1 m_j^1)}{2}$. This nested architectural design can be instantiated as repeated levels of factorized diffusion, which is a promising way to handle multiscale scenes. As a proof of concept, we experiment on the shape 3D dataset [5] with a 2-level hierarchy. We first visualize each level's region mask in Figure 16. We observe that the first level generates a coarse segmentation, based on which, second-level factorized diffusion generates fine segmentations of 3D shapes. Figure 17 provides a more direct visualization of partitions at each level through a 3-class mapping.

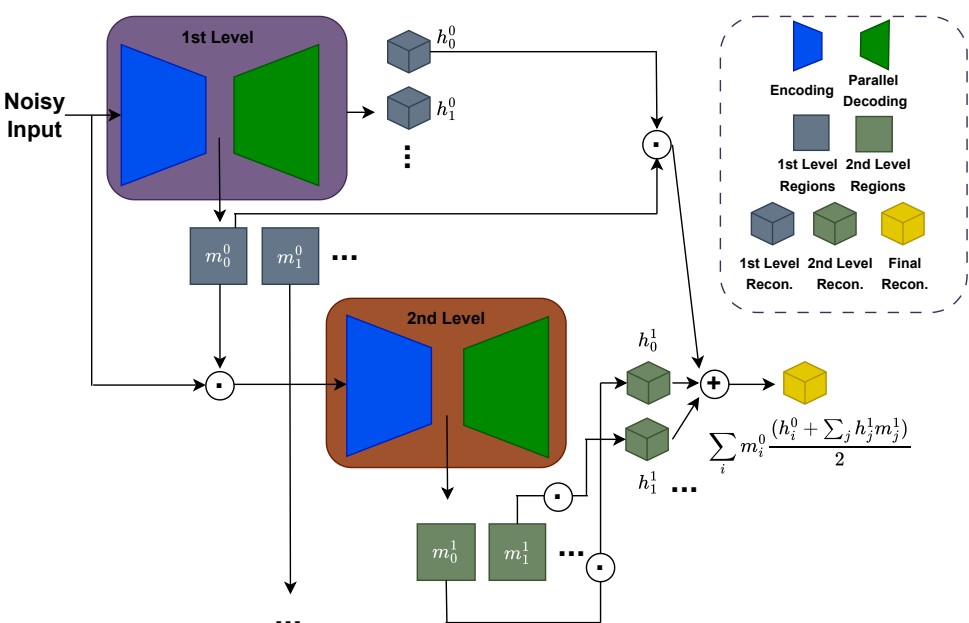

Figure 15: **Hierarchical factorized diffusion architecture.**

## A.2   Additional Segmentation Results

We show more segmentation results for Flower, CUB, FFHQ, CelebA and ImageNet. As shown in Figures 19, 20, 21, 22, and 23, our method consistently predicts accurate segmentations for real image inputs.

## A.3   Additional Generation Results

We show more generation results for Flower, CUB, FFHQ, and ImageNet (classes: flamingo, water buffalo, garbage truck, and sports car). As shown in Figures 24, 25, 26, and 27, our method consistently produces images and masks with high quality.

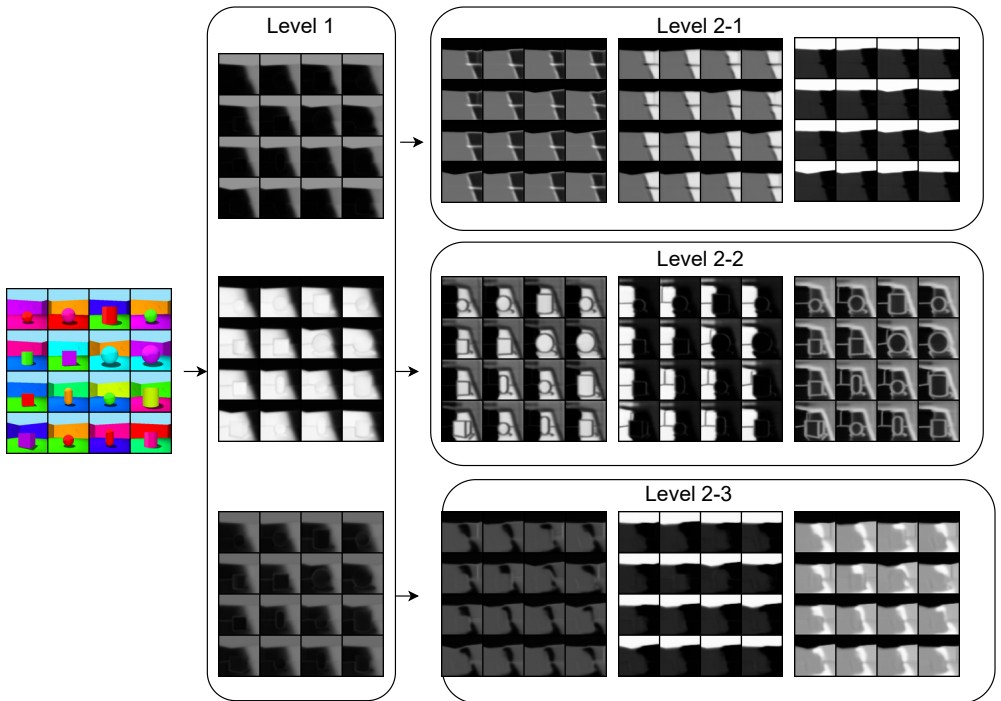

Figure 16: **Mask factorization for each level.** *Level 1:* visualization of each mask channel at the first level. *Level 2-1, 2-2, 2-3:* visualization of each mask channel per branch at the second level.

## A.4 Additional Zero-shot Results on VOC

We provide more segmentation results of 'bicycle', 'chair', 'potted plant' and 'train' in Figure 28.

## A.5 Additional Zero-shot Results on DAVIS

We provide more DAVIS-2017 video segmentation results of 'classic-car', 'dance-jump' in Figure 29.

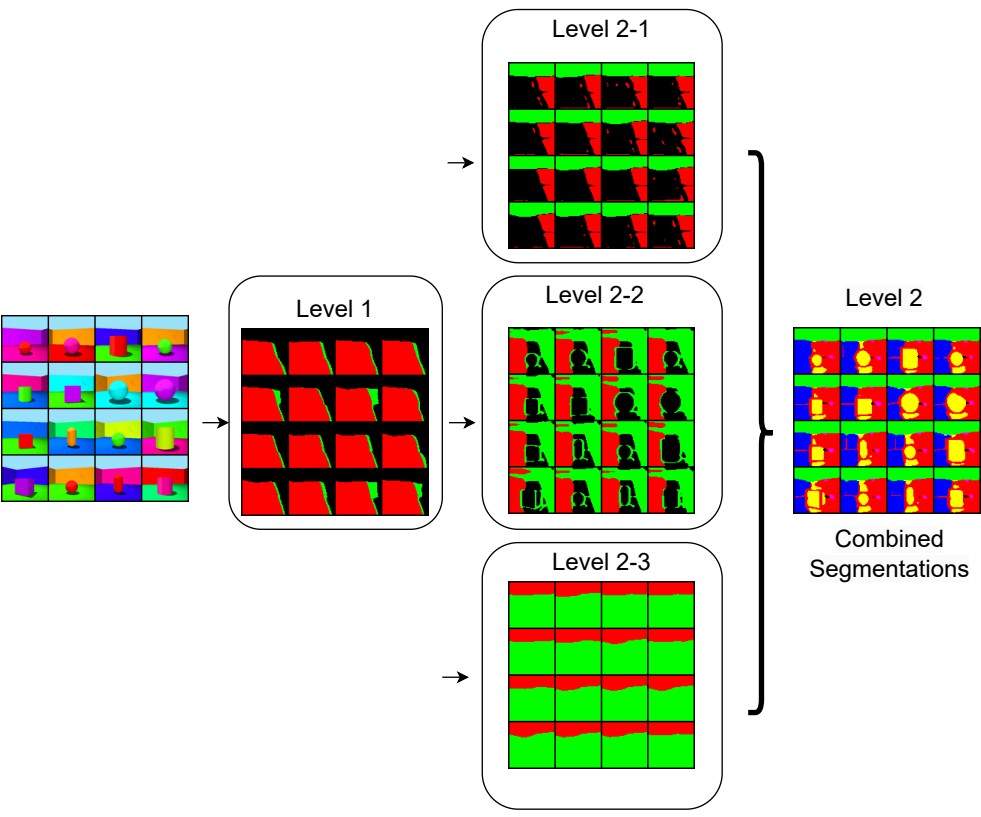

Figure 17: **Segmentations for each level.** *Level 1:* 3-color-coded region assignments at the first level. *Level 2-1, 2-2, 2-3:* 3-color-coded region assignments per branch at the second level. *Level 2 combined segmentations:* 9-color-coded region assignments at the second level.

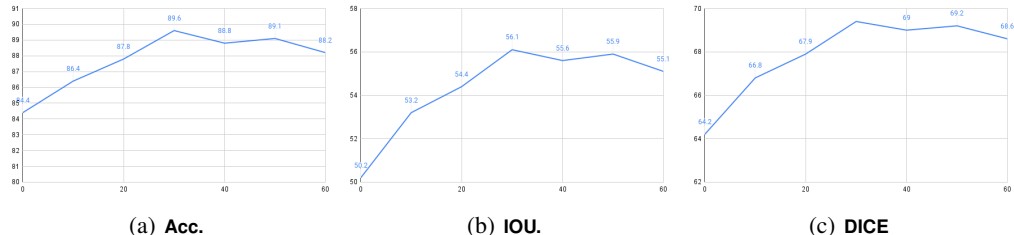

(a) **Acc.**   (b) **IOU.**   (c) **DICE**

Figure 18: **Segmentation results on CUB with** $t \in \{0, 10, 20, 30, 40, 50, 60\}$.

Table 7: We perform class label mapping from ImageNet to VOC, and report zero-shot transfer Accuracy and mIOU per class on VOC validation dataset.

| VOC Class. | ImageNet Class. | Num. of VOC-val Image | Accuracy | mIOU |
|---|---|---|---|---|
| 1:aeroplane | 895:warplane | 136 | 0.82 | 0.57 |
| 2:bicycle | 671:mountain-bike | 108 | 0.79 | 0.47 |
| 3:bird | 94:hummingbird | 168 | 0.83 | 0.58 |
| 4:boat | 814:speedboat | 115 | 0.81 | 0.51 |
| 5:bottle | 907:wine-bottle | 133 | 0.76 | 0.47 |
| 6:bus | 779:school-bus | 114 | 0.73 | 0.54 |
| 7:car | 817:sports-car | 191 | 0.74 | 0.48 |
| 8:cat | 281:tabby | 206 | 0.82 | 0.66 |
| 9:chair | 765:rocking-chair | 175 | 0.75 | 0.64 |
| 10:cow | 346:water-buffalo | 102 | 0.82 | 0.45 |
| 11:diningtable | 532:dining-table | 89 | 0.69 | 0.62 |
| 12:dog | 153:maltese-dog | 204 | 0.82 | 0.67 |
| 13:horse | 603:horsecart | 104 | 0.84 | 0.53 |
| 14:motorbike | 670:motorscooter | 117 | 0.76 | 0.52 |
| 15:person | 981:ballplayer | 584 | 0.77 | 0.46 |
| 16:potted plant | 883:vase | 116 | 0.74 | 0.46 |
| 17:sheep | 348:ram | 89 | 0.84 | 0.64 |
| 18:sofa | 831:studio-couch | 109 | 0.73 | 0.51 |
| 19:train | 466:bullet-train | 126 | 0.76 | 0.56 |
| 20:tv/monitor | 664:monitor | 106 | 0.73 | 0.47 |
| Average | - | - | 0.78 | 0.54 |

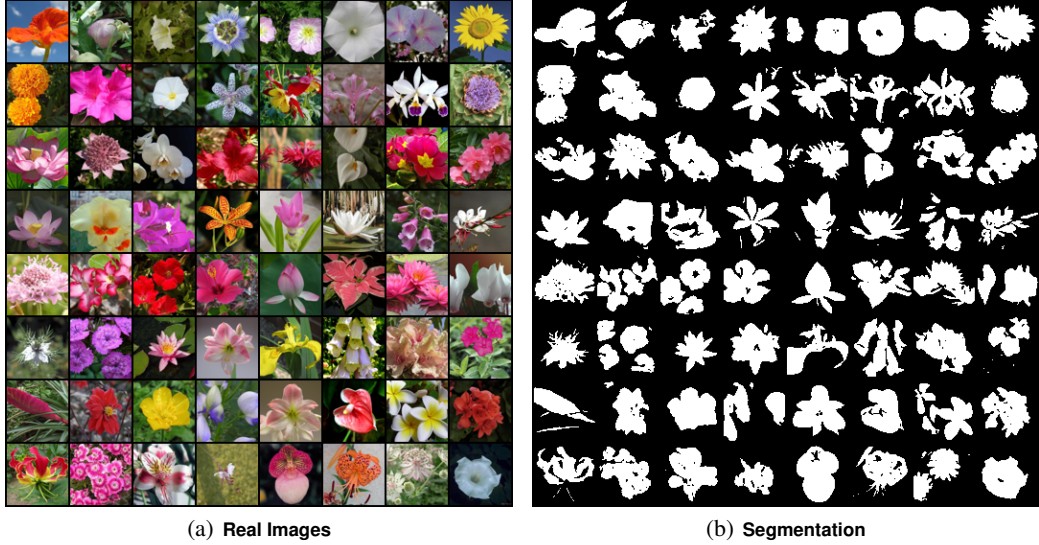

(a) **Real Images**        (b) **Segmentation**

Figure 19: Segmentation on Flower.

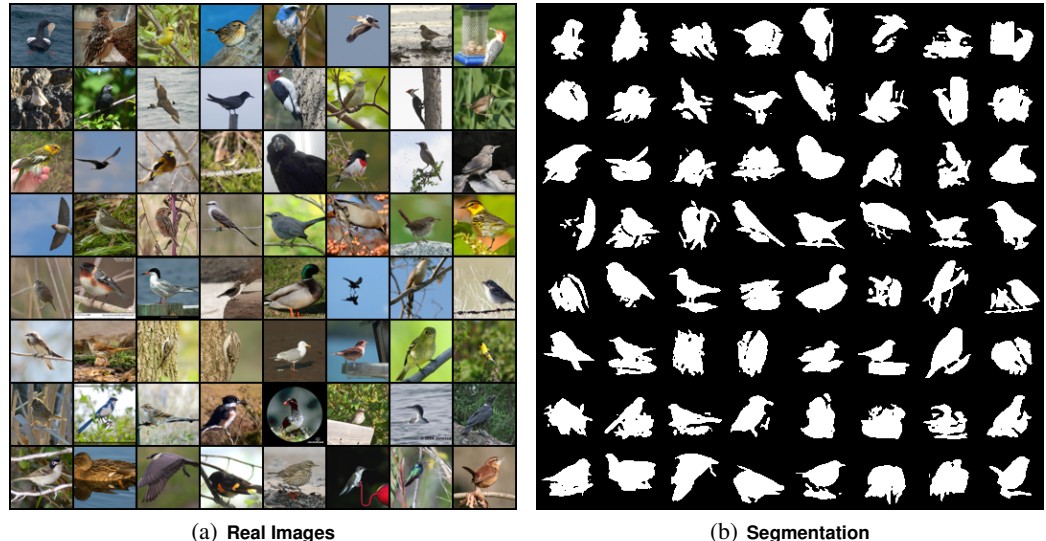

(a) **Real Images**
(b) **Segmentation**

Figure 20: Segmentation on CUB.

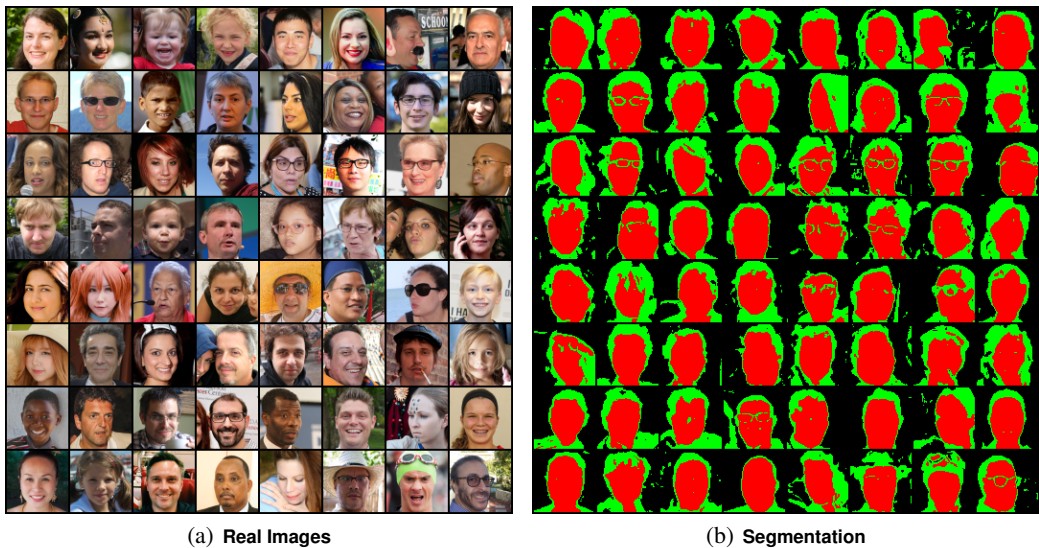

(a) **Real Images**
(b) **Segmentation**

Figure 21: Segmentation on FFHQ.

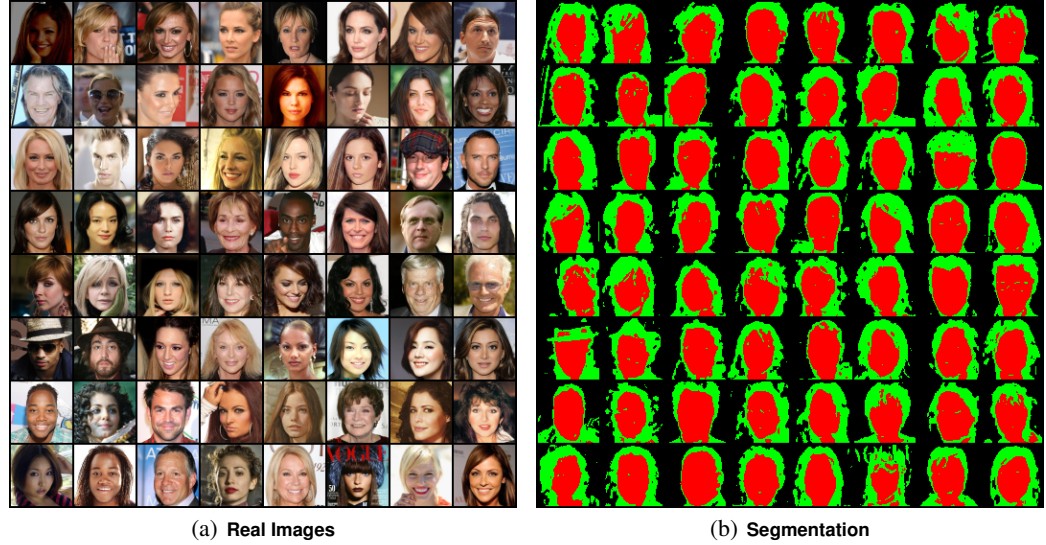

(a) **Real Images**      (b) **Segmentation**

Figure 22: Segmentation on CelebA.

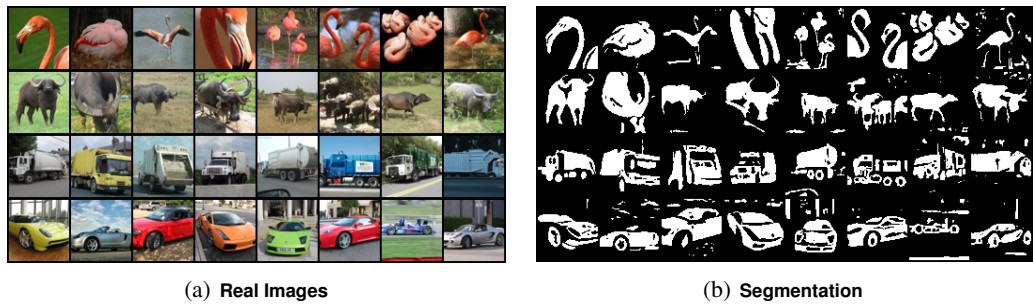

(a) **Real Images**      (b) **Segmentation**

Figure 23: Segmentation on ImageNet.

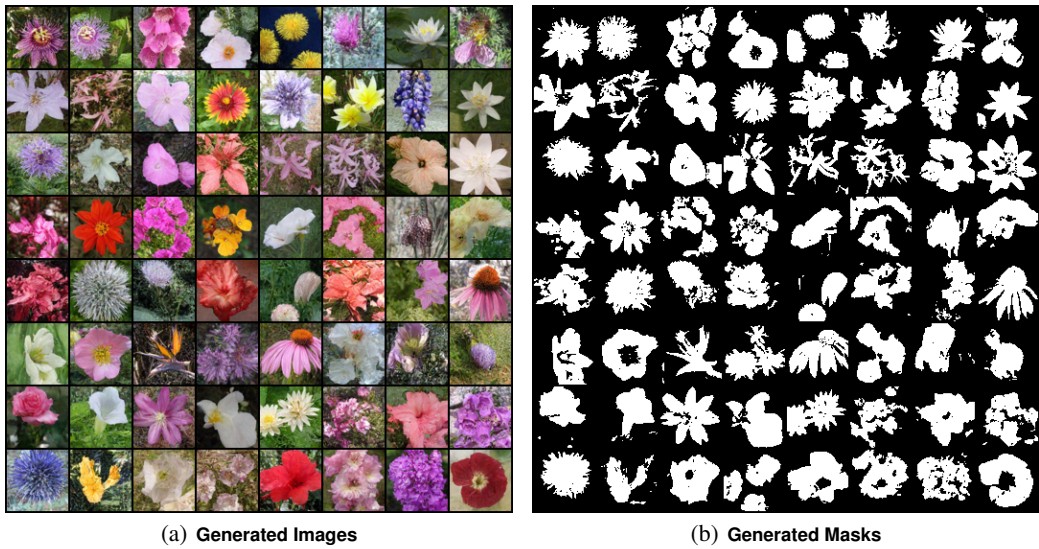

(a) **Generated Images**      (b) **Generated Masks**

Figure 24: Generation on Flower.

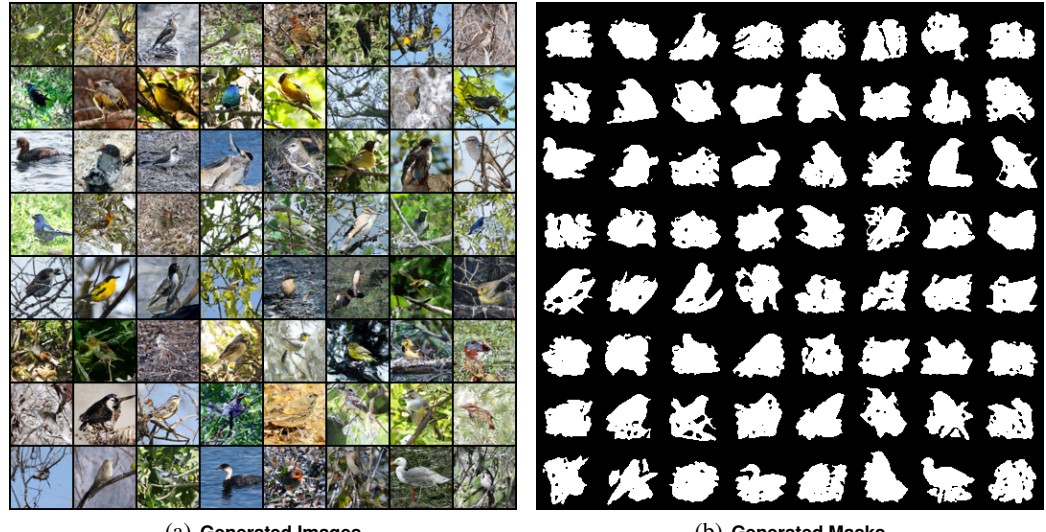

(a) **Generated Images**  (b) **Generated Masks**

Figure 25: Generation on CUB.

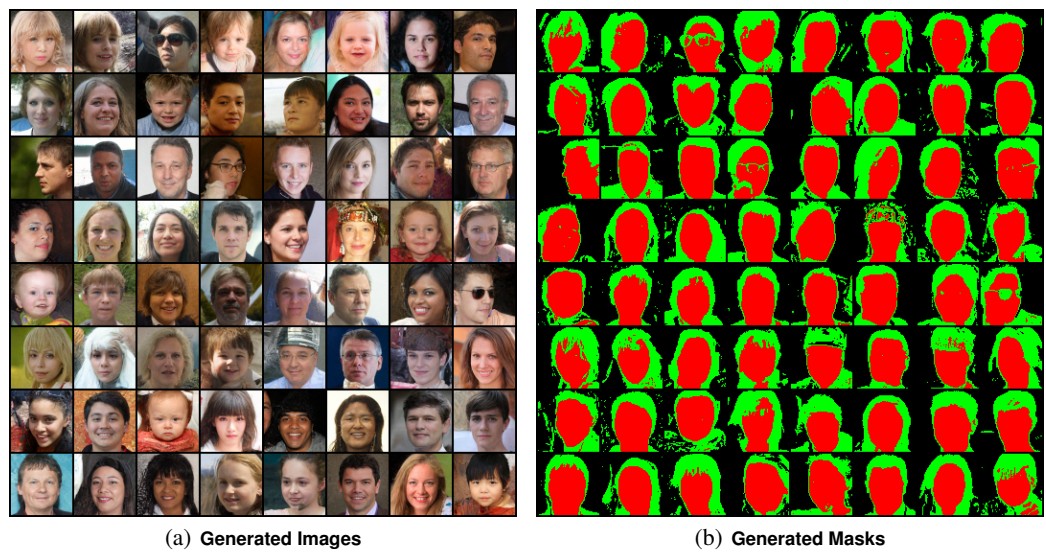

(a) **Generated Images**  (b) **Generated Masks**

Figure 26: Generation on FFHQ.

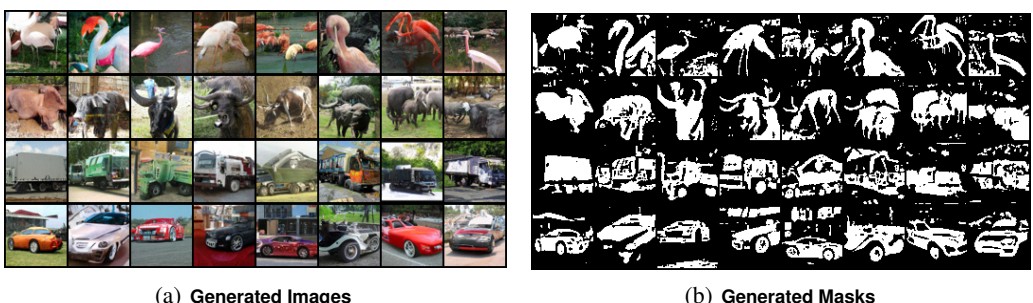

(a) **Generated Images**  (b) **Generated Masks**

Figure 27: Conditional ImageNet generation.

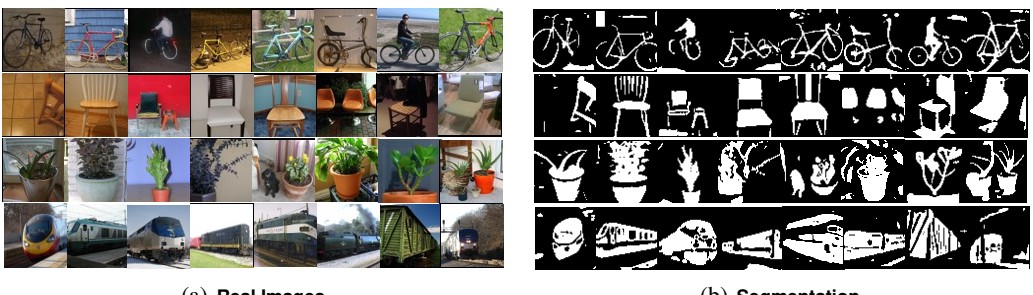

(a) **Real Images**          (b) **Segmentation**

Figure 28: Segmentation on VOC-2012.

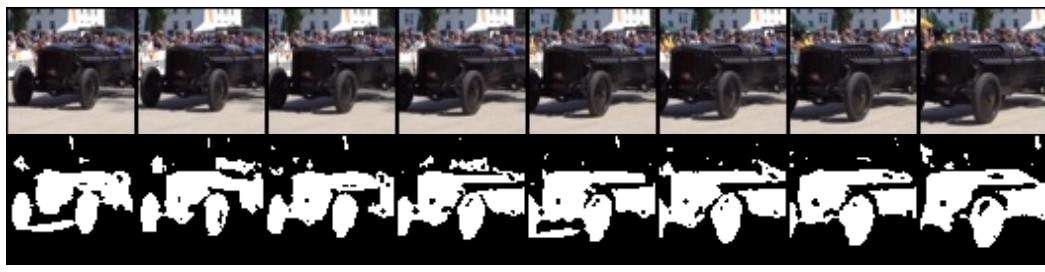

(a) **Frames of 'Classic-car'**

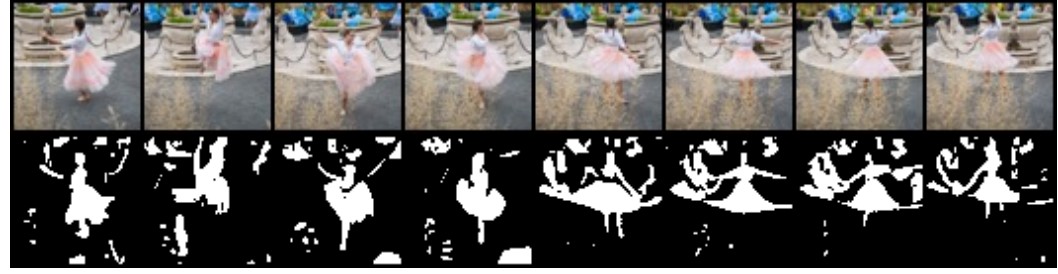

(b) **Frames of 'Dance-jump'**

Figure 29: Segmentation on DAVIS-2017.

