# OpenReview forum: "Factorized Diffusion Architectures for Unsupervised Image Generation and Segmentation"
_NeurIPS.cc/2024/Conference — NeurIPS 2024 poster_

### Official Review · Reviewer_FyWe · 2024-07-03

**Soundness:** 3
**Presentation:** 3
**Contribution:** 3
**Rating:** 5
**Confidence:** 5

**Summary:**

This paper proposes a novel architecture for denoising diffusion probabilistic models (DDPMs) that enables simultaneous unsupervised image generation and segmentation. The key idea is to constrain the denoising network with a structured bottleneck that factorizes the image into regions, which are then denoised in parallel. This architectural design encourages the model to learn meaningful segmentations as a byproduct of training for image generation. The authors demonstrate that their approach can generate high-quality images along with corresponding segmentations, as well as segment real images, without any supervision or additional loss terms. Experiments on multiple datasets show improvements in both image quality and segmentation accuracy compared to baselines.

**Strengths:**

* Novel architectural design that unifies unsupervised image generation and segmentation in a principled way
* Achieves strong results on both tasks without requiring additional loss terms or supervision
* Provides insights into how the model learns to factorize images into regions
* Explores extensions to hierarchical segmentations

**Weaknesses:**

* Limited theoretical analysis:
The paper lacks a rigorous theoretical foundation for why the proposed architecture leads to meaningful segmentations. While empirical results are strong, a deeper analysis of why factorizing the denoising process encourages semantic segmentation would strengthen the contribution. For instance, the authors could explore connections to information bottleneck principles or analyze the gradients flowing through different parts of the network.
* Scope of experiments:
The experiments primarily focus on datasets with relatively simple segmentations (2-3 regions), such as Flower, CUB, and face datasets. While the ImageNet results are promising, they are limited in scope.
The paper lacks experiments on more challenging datasets with complex, multi-object scenes (e.g., COCO, Cityscapes) that would better demonstrate the method's scalability and generalizability.
* Incomplete comparisons:
The paper misses comparisons with some recent, relevant unsupervised segmentation methods, such as PiCIE (CVPR 2021) or STEGO (ICCV 2021). Including these would provide a more comprehensive evaluation of the state-of-the-art.
For the image generation task, comparisons with other recent diffusion model variants (e.g., latent diffusion models) would be valuable to contextualize the improvements.
* Architectural choices and ablations:
The paper does not extensively explore variations in the factorized architecture. For instance, how does performance change with different numbers of parallel decoders or alternative ways of combining their outputs?
More comprehensive ablation studies would help isolate the impact of different components of the proposed architecture.

**Questions:**

* The setting of generating both the images and the segmentation maps is interesting. However, Isn't this setup a little trival? Why we need to generate both contents at the same time? How could this feature help us?
* How sensitive is the method to the choice of number of regions K? Is there a way to automatically determine the optimal K for a given dataset?
* The paper mentions that the method can be extended to hierarchical segmentations. Could you provide more details on how this would work and what challenges might arise?
* Have you explored applying this method to more complex segmentation tasks with many object categories? What modifications might be needed?
* How computationally expensive is the proposed method compared to standard DDPMs, both for training and inference?
* The paper claims the method can be applied to segment novel images with just one forward pass. How does the runtime compare to other unsupervised segmentation methods?
* Some literature that share the similar insights of unifying generation and segmentation can be refered to:
[1] Lai Z, Duan Y, Dai J, et al. Denoising diffusion semantic segmentation with mask prior modeling[J]. arXiv preprint arXiv:2306.01721, 2023.
[2] Li C, Liu X, Li W, et al. U-KAN Makes Strong Backbone for Medical Image Segmentation and Generation[J]. arXiv preprint arXiv:2406.02918, 2024.

**Limitations:**

The authors have adequately addressed the limitations of their work in Section 5, acknowledging that the current results are limited to 2-3 class segmentations and discussing the need for further work on handling more complex scenes. They also mention computational costs as a potential limitation. The paper does not directly discuss potential negative societal impacts, which could be briefly addressed, though the risks seem relatively low for this type of fundamental methodological work.

---

> ### Author Rebuttal · Authors · 2024-08-07
>
> # To Reviewer FyWe
>
> **Q: Limited theoretical analysis.**
>
> A: Our system could potentially learn to assign the entire image to a single mask, leaving all other masks empty. In doing so, it would essentially fall back to being equivalent to a standard DDPM UNet architecture. All synthesis would occur as a result of a single run of the decoder. However, if the system learns to split the image into different regions, it gets to use multiple runs of the decoder in order to synthesize the denoised result. There is a computational advantage (using more decoder runs) that the system can leverage to better denoising, if it can learn how to partition the problem into parallel subtasks. This is the fundamental driver behind the denoising loss encouraging learning of region partitions. We agree that a quantitative analysis of these dynamics would be valuable.
>
> **Q: Scope of experiments and Incomplete comparisons; Have you explored applying this method to more complex segmentation tasks with many object categories?**
>
> A:**Regarding dataset selection**, our work is at the stage of a new architectural design for diffusion-based segmentation and generation, with 2 or 3 class segmentation results demonstrating the improvement across multiple datasets, scaling up to ImageNet.
>
> Training unconditioned DDPM from scratch for complex domains, e.g., COCO or PASCAL, is challenging, even for the single goal of high-quality image generation. Simultaneous generation and segmentation introduces another level of difficulty. Our work proposes and validates new ideas on object-centric datasets. Subsequent efforts, and more computational resources, will be needed to scale up our techniques to larger and more complex datasets.
>
> Moreover, our work is itself on the path of scaling the segmentation capabilities of DDPMs in comparison to prior work. Consider the evolution from: DatasetDDPM (ICLR'2022) introduced the idea of using a **pre-trained** DDPM to extract segmentations in a **supervised** manner, validated on the datasets up to the scale of **CelebA**, to: Ours (submission'2024) trains a factorized diffusion model **from scratch** to accomplish both segmentation and generation **simultaneously** in an **unsupervised** manner, validated on the larger scale ImageNet dataset with **more complex scenes**.
>
> **Regarding baseline selection**, our goal is to perform both unsupervised segmentation and image generation in a unified framework. This is an entirely novel combination of capabilities; our work is an initial proof-of-concept, not an attempt to achieve state-of-the-art unsupervised segmentation results. The relevant competing baselines mainly lie in the scope of generative models for unsupervised segmentation. Our approach is able to achieve better unsupervised segmentation results across multiple datasets. We can also achieve improved generation quality (over standard DDPM) as shown in Experimental Section 4.1. Simply putting the state-of-the-art unsupervised segmentation methods in comparison with our approach ignores the more challenging setting in which we operate (requiring that we are also able to generate images), as well as the fact that our system improves generation quality.
>
> **Q: Architectural choices and ablations.**
>
> A: We do have a more systemic investigation in different architectural choices in Appendix Section 3: reorganizing our architectural design to support hierarchical mask factorization in place of a flat set of $K$ regions. Additionally, during the rebuttal phase, we enlarge the flat set of $K$ regions to 5 for CLEVR dataset, as shown in Figure 2 of the rebuttal PDF. We also show the results with K=2 were found to be less satisfactory than K=3 for both segmentation and generation in Table 1 (rebuttal PDF).
>
> **Q: Why need to generate both contents at the same time?**
>
> A: We only train the model once and achieve both targets. Doing both together actually improves generation quality, while yielding the ability to segment as a bonus. Alternatively, through the lens of a traditional computer vision task, we learn to segment in an unsupervised manner and get the ability to generate as a bonus.
>
> **Q: How sensitive to the choice K?**
>
> A: We are defining the maximum number of regions the model can use for factorization.  There is no requirement that it utilize all of these mask components. For example, it could learn to only use two mask channels out of three. Or it could predict every pixel as belonging to the same mask channel, thereby collapsing back to the vanilla UNet as a base case. The fact that the model prefers to learn something nontrivial is related to the actual structure of the data; nothing forces it to use all K channels.
>
> **Q: More details on hierarchy segmentation.**
>
> A: We have detailed the investigation in hierarchy segmentation in Appendix A.3.
> We formulate a hierarchical factorized diffusion architecture to progressively refine segmentation results from a coarse initial prediction to a fine, detailed final segmentation. Our initial investigation into hierarchical extensions suggests a promising future path towards handling complex scenes.
>
> **Q: How computationally expensive for training and inference, compared with DDPMs?**
>
> A: Given the nice property of weight sharing scheme in parallel decoding, the only additional weights are from our mask generator. We have an efficient batching implementation for decoding, which makes training and sampling speed comparable to standard DDPMs. As for segmentation, the inference speed is the same as a single forward pass in a standard DDPM.
>
> **Q: Runtime compared to other unsupervised segmentation methods?**
>
> A: If both use the same UNet encoder-decoder architecture, the inference complexity is the same. The parallel decoding scheme and diffusion process does not affect the efficiency of segmentation inference speed.
>
> **Q: Some literature can be referred to.**
>
> A: Thanks for the suggestion. We will include a discussion in the final version.

---

> > ### Comment · Reviewer_FyWe · 2024-08-10
> >
> > > Q: Why need to generate both contents at the same time?
> > >> A: We only train the model once and achieve both targets. Doing both together actually improves generation quality, while yielding the ability to segment as a bonus. Alternatively, through the lens of a traditional computer vision task, we learn to segment in an unsupervised manner and get the ability to generate as a bonus.
> >
> > Regarding this point, is there any evidence (theoretically or empirically) that could support this claim? It seems not that straightforward.
> >
> > Thank you!
> >
> > Best,

---

> > > ### Author Response · Authors · 2024-08-13
> > > **To Reviewer FyWe**
> > >
> > > **Q: Is there any evidence (theoretically or empirically) that could support this claim [doing both together actually improves generation quality]?**
> > >
> > > A: Yes, Table 5 (top) gives empirical evidence for precisely this claim. Our system, which is a generation architecture that internally performs segmentation, generates higher quality images (lower FID) than a standard DDPM baseline. For each dataset in Table 5, we train from scratch each system (ours and the DDPM baseline) in a fully unsupervised manner driven by the denoising objective alone. Our system and the DDPM baseline have comparable design motifs and similar parameter counts, with the distinction that, as described in Sections 3.1 and 3.2, ours adds a mask generator module and partitions synthesis into parallel decoding pathways. This structural change to the architecture significantly improves FID across all datasets (e.g., 13.35 to 10.79 for FFHQ-128 and 7.02 to 6.54 for ImageNet-64).
> > >
> > > Thus, our system is a denoising diffusion model with **improved generation quality** over a baseline DDPM, and is trained in the exact same *fully unsupervised manner*. If someone were only interested in high quality generation, our system would be preferable to the standard DDPM. Beyond this, our system **produces segmentation as a bonus**; segmentation can be read from the internal bottleneck state (region masks) of our architecture. Alternatively, someone interested in learning to segment in an unsupervised manner could view the image generation capability of our system as a bonus. As shown in Figure 1 (a)(b), we only train our model once and achieve both targets (image generation, segmentation). The fact that our architecture improves generation quality makes it a candidate to serve as the basis for future diffusion-based foundation models.
> > >
> > > Viewed in a broader context, it is perhaps not too surprising that network architecture design can have a strong influence on learning and on prediction or output quality. Convolutional neural networks and attention mechanisms in transformers are two examples of imbuing neural architectures with domain-relevant structure. We are imbuing domain-relevant structure at a more macro scale, in the form of parallel synthesis pathways that are a match to a compositional model of images.

---

### Official Review · Reviewer_vzUs · 2024-07-11

**Soundness:** 2
**Presentation:** 3
**Contribution:** 2
**Rating:** 4
**Confidence:** 4

**Summary:**

The authors propose a model for simultaneous unsupervised semantic segmentation and image generation based on denoising diffusion probabilistic models. The methodology alters the architecture of a typical DDPM by conditioning the decoder on the outputs of a mask generator.

**Strengths:**

In general, the paper is well-written and technically sound.

The experimental results surpass the state-of-the-art on the datasets they were shown and the results seem promising.

**Weaknesses:**

The abstract lacks context/motivation. The conclusions and experiments lack discussions of the limitations of the proposed approach.

The authors propose a simple solution to a complex problem (unsupervised segmentation) through altering the architecture of a DDPM by adding an additional decoder to generate masks in an unsupervised manner and conditioning the DDPM’s original decoder on the
outputs of the mask generator. However, the explanation of the model is unclear:
Initially, 𝑚𝑘 is introduced as the output of the mask generator after applying the softmax activation function, for each of the k channels. As such, 𝑚𝑘 should have dimensions width x height x 1, where width and height are the dimensions of the original images. Then, it is stated
that 𝑚𝑘 is concatenated with ℎenc (constituted by several outputs of the encoder provided to the decoder as skip connections) and ℎ𝑚𝑖d (latent vector) and provided as input. It is unclear how these variables of different dimensions are concatenated. In the manuscript, it says “We downsample 𝑚 accordingly to the same resolution as ℎ𝑚𝑖d and ℎenc at different stages” (line 157), however it is still unclear how exactly this downsampling step is achieved. Does 𝑚 represent the outputs of each layer of the mask generator and is it provided as input to the decoder as skip connections? If so, how is 𝑚𝑘 obtained from 𝑚? Do the outputs of all layers of the mask generator possess exactly 𝑘 channels? An example highlighting the dimensions of 𝑚𝑘, ℎ𝑚𝑖d and ℎenc and how these are concatenated would be helpful.

The experiments are a limited when it comes to the datasets used to evaluate the model for semantic segmentation. The only two datasets used for over 2 classes (distinguishing more than background and foreground) were face datasets, where all images are focused on the face and present relatively low variability. It is unclear how the method would work with over 3 classes and on datasets with higher variability such that not all objects are present in all images. Furthermore, even when only two classes are used, to distinguish background from foreground, the datasets seem to be limited, with low variability in backgrounds, making it difficult to assess the model’s results. The results on the CUB dataset, whose backgrounds present more variability and the regions with objects of interest (birds) are smaller, show difficulties segmenting the birds from the backgrounds, especially in the presence of branches (Fig 8). In the ImageNet experiment, it would also be interesting to see what would happen if 5 classes (4 objects + background) were used (k=5 in the softmax layer) rather than just two.
Would the network be able to learn to distinguish certain objects or would it still only be able to separate background from foreground? As it stands, it becomes difficult to assess the proposed network’s strengths and whether it is truly capable of capturing semantic information or whether it just separates background from foreground without requiring more in-depth semantic information about the nature of the objects.

The paper lacks a discussion of the limitation of the methods.

**Questions:**

see above the comments about the concatenation of ℎ𝑚𝑖d and ℎenc.

**Limitations:**

yes.

---

> ### Author Rebuttal · Authors · 2024-08-07
>
> # To Reviewer vzUs
>
> **Q: The abstract lacks context/motivation. The conclusions and experiments lack discussions of the limitations of the proposed approach.**
>
> A: In the current abstract, we provide a clear introduction to the challenges and motivations driving our research. The abstract begins by discussing the limitations of supervised deep learning, specifically the reliance on large amounts of annotated data, which is a well-known barrier in computer vision tasks. We explicitly mention the goal of achieving unsupervised image generation and segmentation, addressing the significant problem of high annotation costs in supervised learning. The abstract succinctly outlines our approach—a denoising diffusion model with a computational bottleneck—which is a novel strategy for tackling the dual tasks of generation and segmentation simultaneously. It concludes by highlighting the success of our model in achieving high-quality image synthesis and segmentation across multiple datasets, reinforcing the effectiveness of our approach.
>
> The conclusion and experiment sections of the paper do acknowledge and discuss limitations, which are crucial for a balanced understanding of the proposed method. The conclusion acknowledges that the work is at an early stage of architectural design, focusing on simpler class scenarios to establish a proof of concept before tackling more complex tasks. We discuss that the experiments are performed primarily on two to three class scenarios, recognizing these as initial limitations with plans to extend the approach to higher resolutions and more complex segmentation tasks in future work.
> The experiments highlight the need for further exploration of scalability and generalization to more diverse datasets and tasks. This is particularly important for understanding how the model performs in real-world applications beyond the current scope. The conclusion outlines future research directions aimed at addressing these limitations, such as investigating hierarchical extensions and handling more complex scenes.
>
> **Q: The explanation of the model is unclear .**
>
> A: This question is about Eq.(5), which is not our proposed design. In line 158-160, "However, such a design significantly modifies (e.g., channel sizes) the original U-Net decoder architecture. Moreover, conditioning with the whole mask representation may also result in a trivial solution that simply ignores region masks." As such, there's no concatenation of mask and $h_enc$ in our proposed parallel decoding scheme. Refer to Eq.(6) and Eq. (7) for the description of our design.
>
> We again clarify the design as depicted in Figure 2: $h_{mid}$ (latent features) is directly passed to the decoders following conventional DDPM without concatenation.
> $m_k$ denotes the $k-th$ channel of the mask generator **final** output $m$. The mask $m_k$ is used at each decoding layer as part of skip connections. It provides region-specific information that helps the decoder refine outputs by distinguishing between different semantic regions.
> As shown in Figure 2, we adopt **channel-wise masking**: Each channel of $m$, denoted as $m_k$, is applied to $h_{enc}$ through **element-wise multiplication**.
> We downsample $m_k$ using bilinear interpolation to align its spatial dimensions with $h_{enc}$ at different stages. Only element-wise multiplication is used with consistent spatial dimensions, allowing integration across the network's architecture without significantly changing the channel dimension of conventional DDPM.
>
> Q: **The experiments are a limited when it comes to the datasets used to evaluate the model for semantic segmentation.**
>
> A: Training unconditioned DDPM from scratch for complex domains, e.g., COCO or PASCAL, is challenging, even for the single goal of high-quality image generation. Simultaneous generation and segmentation introduces another level of difficulty. Our work proposes and validates new ideas on object-centric datasets. Subsequent efforts, and more computational resources, will be needed to scale up our techniques to larger and more complex datasets.
>
> Moreover, our work is itself on the path of scaling the segmentation capabilities of DDPMs in comparison to prior work. Consider the evolution from: DatasetDDPM (ICLR'2022) introduced the idea of using a **pre-trained** DDPM to extract segmentations in a **supervised** manner, validated on the datasets up to the scale of **CelebA**, to: Ours (submission'2024) trains a factorized diffusion model **from scratch** to accomplish both segmentation and generation **simultaneously** in an **unsupervised** manner, validated on the larger scale ImageNet dataset with **more complex scenes**.
>
> As for the segmentation results, we provide a supervised result as a reference point against which to compare unsupervised methods using the same architecture. One would not expect any unsupervised method, limited to training on the same data, to be able to match the supervised method's performance. Our method, which is unsupervised, outperforms the unsupervised baseline (DatasetDDPM-unsup) based on the same UNet architecture.
>
> It is the case that we want to explore much larger K for ImageNet in the future, as doing so will be necessary for segmenting complex scenes. A challenge is implementing K parallel pathways efficiently for large K; note that training is already expensive as everything takes place within a diffusion model.

---

### Official Review · Reviewer_FkB4 · 2024-07-12

**Soundness:** 3
**Presentation:** 3
**Contribution:** 2
**Rating:** 4
**Confidence:** 4

**Summary:**

The paper introduces a novel neural network architecture capable of simultaneous image generation and segmentation in an unsupervised manner, eliminating the need for pre-labeled data. The core concept involves training the network to dissect an image, clean individual sections, and reassemble them. Notably, the system can decipher the image content solely through the analysis of these sections. The authors demonstrate the effectiveness of their unsupervised model for real-image segmentation. This work presents a promising approach for generating realistic images and understanding their content with minimal reliance on labeled data.

**Strengths:**

- Clear and concise presentation
- Experimental results validate the proposed method's effectiveness

**Weaknesses:**

- Limited discussion of related work: The paper lacks a comprehensive review of relevant research, hindering the assessment of genuine novelty. It's crucial to mention existing works like DatasetGAN (utilizing inner features of trained GANs for segmentation), Diffuse, Attend, and Segment (high-quality zero-shot segmentation with attention maps), DINO, and FeatUP (representation utilization in generative models for downstream tasks).

**Questions:**

- How does the proposed method differ from the mentioned related works? What are the key advantages it offers?
- Are the results quantitatively compared with existing segmentation methods (e.g., DatasetGAN, etc.)?

**Limitations:**

The ethical implications of using generative models to create harmful content should be addressed.

---

> ### Author Rebuttal · Authors · 2024-08-07
>
> # To Reviewer FkB4
> **Q: Limited discussion of related work; How does the proposed method differ from the mentioned related works? What are the key advantages it offers?**
>
> A: Thanks for the suggestion. We will incorporate this discussion into our related work section to provide a clear picture of how our approach diverges from existing methods:
>
> - DatasetGAN has utilized intermediate GAN features to perform segmentation, demonstrating the potential of leveraging generative models for downstream tasks. However, our approach differs by directly integrating segmentation within the diffusion process, eliminating the need for separate feature extraction and potentially offering more coherent results.
>
> - Similarly, Diffuse, Attend, and Segment employs attention mechanisms for zero-shot segmentation, providing high-quality segmentation through attention-driven methods. Our factorized diffusion architecture offers an alternative by producing segmentation masks as an intrinsic part of the denoising process, potentially yielding more integrated and coherent segmentation results.
>
> - In the realm of self-supervised learning, DINO showcases the power of representation learning through knowledge distillation, applicable to tasks like segmentation. Our approach aligns with this goal of versatile representation learning but achieves segmentation directly within the generative model, eliminating the need for additional supervision or distillation processes.
>
> - FeatUP highlights the adaptability of generative model representations for downstream tasks. While FeatUP focuses on broad applicability, our method targets segmentation with a specifically designed architecture, achieving simultaneous image generation and segmentation in a unified framework.
>
> In comparison with prior work, we are offering an entirely new approach to solving an end task (segmentation) using unsupervised generative learning: (1) Specify an architecture whose bottleneck representation encodes the solution to the end task, and whose decoder's computational structure constrains synthesis of data from that representation. (2) Train the system end-to-end, from scratch, for generation alone. (3) Read off the bottleneck features as the solution.
>
> **Q: Are the results quantitatively compared with existing segmentation methods (e.g., DatasetGAN, etc.)?**
>
> A: Both DatasetGAN and DatasetDDPM are not unsupervised approaches like ours. They require annotations and train in a few shot manner. In the paper, we compared with DatasetDDPM, which is a more suitable baseline. As shown in Table 3 and 4, our method outperforms DatasetDDPM by a large margin under the same unsupervised setting.
>
> **Q: The ethical implications of using generative models to create harmful content should be addressed.**
>
> A: We are committed to ensuring that the development and application of generative models are guided by ethical principles. We will incorporate the discussion in the final version.

---

### Official Review · Reviewer_9WST · 2024-07-12

**Soundness:** 4
**Presentation:** 4
**Contribution:** 3
**Rating:** 6
**Confidence:** 3

**Summary:**

The paper proposed a structural modification of a DDPM which causes it to learn a decomposition of images into regions. This factorization enables unsupervised segmentation and simultaneously improves the quality of the generated images. The method is evaluated on various datasets, demonstrating its effectiveness in both tasks.

**Strengths:**

- Evaluations on 5 datasets.
- Consistently good performance relative to other unsupervised methods.
- Conceptually simple but powerful design.
- Ablation on encoder design details.
- Zero-shot segmentation shown on 2 additional datasets.

**Weaknesses:**

- Experiments only cover 2-3 class scenarios.
- ImageNet results only on downsampled 64x64 images.

**Questions:**

- In addition to DiffuMask, please consider discussing DiffSeg (https://arxiv.org/pdf/2308.12469), which like DiffuMask, relies on a Stable Diffusion model, but unlike it, does output masks explicitly.
- Is the proposed scheme specific to diffusion? Could it be applied in other autoencoder settings?
- What happens if you set K to higher values, despite not needing the additional classes? e.g. K = 4, 5 for binary segmentation?
- You mentioned that K = 3 was helpful for binary segmentation. How much worse were the results with K=2?

**Limitations:**

Limitations are discussed explicitly in the paper. No concerns about negative societal impact.

---

> ### Author Rebuttal · Authors · 2024-08-07
>
> # To Reviewer 9WST
>
> **Q: Experiments only cover 2-3 class scenarios.**
>
> A: Our work is at the stage of a new architectural design for diffusion-based segmentation and generation, with 2 or 3 class segmentation results demonstrating improvements across multiple datasets, scaling up to ImageNet.
>
> Training unconditioned DDPM from scratch for complex domains, e.g., COCO or PASCAL is challenging, even if limited to the goal of high-quality image generation. We are addressing an additional challenge of simultaneously generating a latent representation of segmentation.
>
> Our work proposes and validates new ideas on object-centric datasets. We believe there is a path toward scaling our method to more complex data; e.g., see the Appendix for our effort towards hierarchical segmentation. Fully exploring the new technique we have introduced will require follow-up papers.
>
> **Q: ImageNet results only on downsampled 64x64 images.**
>
> A: This choice is due to computational constraints and the need to validate our method's feasibility at a lower resolution before potentially scaling to higher resolutions. Training models on full-resolution (e.g., 256 $\times$ 256) ImageNet images would require significantly more computational resources and time. Resolution is an aspect of the system orthogonal to our architectural innovation.
>
> **Q: In addition to DiffuMask, please consider discussing DiffSeg, which like DiffuMask, relies on a Stable Diffusion model, but unlike it, does output masks explicitly.**
>
> A: Thanks for the suggestion. We will add the discussion about DiffuMask in the revised draft.
>
> **Q: Is the proposed scheme specific to diffusion? Could it be applied in other autoencoder settings?**
>
> A:  While the details of our proposed scheme are tailored for diffusion models, our core principles could potentially be adapted to other autoencoder architectures, e.g., variational autoencoders (VAEs) or masked autoencoders (MAEs):
> - Bottleneck Design: The use of a structured bottleneck could be implemented in autoencoders to encourage learning of meaningful latent representations that aid in segmentation.
> - Parallel Decoding: The idea of parallel decoding for different segments could be utilized in the decoder portion of any autoencoder to encourage factorizing the reconstruction task into independent subtasks.
>
> **Q: What happens if you set K to higher values, despite not needing the additional classes? e.g., K = 4, 5 for binary segmentation?**
>
> A: $K$ is the maximum number of regions the model may use; it could learn fewer. Setting K to higher values may allow for some additional flexibility during training, as the model learns how to partition images into regions. Setting K much larger than necessary simply wastes compute as the model learns to leave some region maps empty (unused). During the rebuttal phase, we experiment with enlarging the flat set of $K$ regions to 5 for the CLEVR dataset, as shown in Figure 2 (rebuttal PDF). The model still performs foreground-background segmentation adequately by ignoring extra classes if not present.
>
> **Q: You mentioned that K = 3 was helpful for binary segmentation. How much worse were the results with K=2?**
>
> A: For binary segmentation, we found setting $K=3$ rather than $K=2$ to assist training, with learned regions emerging as foreground, background, and a contour or transition between the two, as shown in Figure 1 of the rebuttal PDF. Also, as shown in Table 1 in the rebuttal material, the results with K=2 were less satisfactory for both segmentation and generation. With the goal of handling more regions and multiscale structure, we believe a more promising future investigation is to reorganize our architectural design to support hierarchical mask factorization in place of a flat set of $K$ regions, as shown in Appendix Section A.3.

---

> > ### Comment · Reviewer_9WST · 2024-08-12
> >
> > Thank you for thoroughly responding to my questions. I understand the concerns about computational resources -- mentioning this explicitly in the paper would be helpful. Similarly, the fact that the general idea is potentially applicable to other architectures might warrant a short comment in the discussion section.
> >
> > Regarding K=2, I was surprised to see the magnitude of the impact on the metrics. Do you have an intuitive explanation for this? Is it the case that with K=2 the shapes of the masks do not adhere well to the contours of the underlying objects, which K>2 helps to alleviate?

---

> > > ### Author Response · Authors · 2024-08-13
> > > **To Reviewer 9WST**
> > >
> > > We will update our discussion to include computational requirements and potential applicability to other autoencoder frameworks.
> > >
> > > **Q: Regarding K=2 and impact on the metrics**
> > >
> > > The choice of K not only defines an architectural structure for inference, but one that comprises the model during training; K is the maximum number of components that the model may utilize at any point in training or inference. We train from scratch, so at the start of training, the assignment of pixels to components will be a result of random initialization. Training via gradient descent must find a path from this random configuration to a parameter configuration which yields structured region masks. It might be helpful to have more components (more masks) to utilize during training in order to smooth the optimization landscape, even if by the completion of training not all masks are needed. An analogy is overparameterization benefiting neural network training, even though networks are subsequently amenable to pruning. Consistent with this hypothesis is the observation that our K = 3 experiments for objects vs background datasets show the third component emerging as a contour or transition between the object and background (Figure 1 in rebuttal PDF). To provide further analysis, we will add visualization of region mask evolution over the duration of training to the Appendix.

---

### Author Rebuttal · Authors · 2024-08-07

# General Reply

We thank the reviewers and answer specific questions in individual responses to each review below. We provide some general clarifications here, as well as specific clarifications below.

Our goal is both image segmentation and image generation, learned simultaneously and in an entirely unsupervised manner.  Our system resembles a standard DDPM, but innovates on the neural network architecture inside that DDPM.  Our novel network architecture is designed around heterogeneous subcomponents which have the emergent effect of encouraging the DDPM to factorize the denoising process into parallel subproblems and allow us to easily examine the learned factorization.  For images, it happens that the natural factorization, simply as a consequence of the structure of the data, corresponds to region segmentation.  Thus, as a result of learning to generate (denoise) with this particular network architecture, we also learn to segment.

Once trained, our system can operate in two modes: segment an arbitrary input image, or generate an image from random noise.  Segmenting a novel input image is fast (single forward pass of a UNet, comparable in speed to any other image segmentation network); generation is slow (many steps of a reverse diffusion process, just as in a standard DDPM).

Our work represents the first example of a new strategy for leveraging diffusion models to learn other latent information in an unsupervised manner: constrain the generation (denoising) architecture to have a computational structure and representation bottleneck that reveals that desired latent information.  In our case, the latent information is image segmentation, the bottleneck is feature tensors restricted to region masks, and the computational structure is parallel pathways processing those masked feature tensors.  If this recipe proves to be general, it should inspire a new strategy for the architectural design of future foundation models.

---

### Comment · Area_Chair_afhb · 2024-08-09
**Rebuttal is online - please respond**

Dear Reviewers,

Authors carefully prepared their rebuttal - trying to address the concerns you have raised. Please check the rebuttals and join the discussion about the paper.

Regards,

---

### Decision · Program_Chairs · 2024-09-25

**Decision:**

Accept (poster)

**Comment:**

Authors present an unsupervised diffusion model that can not only
generate images but also provide segmentations, provided the number of
masks required as input. The approach and the architecture seems quite
novel as also stated by some reviewers.

The main concern raised during the review was that the extent of
experiments could be larger. Authors could have done experiments with
higher number of classes, full resolution images and more
datasets. During the rebuttal authors note that the main issue is the
computational cost. Given the number of experiments and the ablation
study, I tend to agree with the authors. Some reviewers also agree with
this.

The last reviewer also mentions theoretical insights are not
provided. Authors' response, while not answering it fully, provided
insights. This reviewer gave a BA for this paper.

There are two reviewers who gave BR to this article. Their main
concers were about experiments and comparisons. The second BR
reviewer also mentions clarity and motivation. Authors did their best
to answer these concerns during the rebuttal. Unfortunately, reviewer
did not engage in a discussion. However, reading the rebuttal, I think
authors response were valuable.